# Group Distributionally Robust Machine Learning under Group Level Distributional Uncertainty

## Abstract

The performance of machine learning (ML) models critically depends on the quality and representativeness of the training data. In applications with multiple heterogeneous data generating sources, standard ML methods often learn spurious correlations that perform well on average but degrade performance for atypical or underrepresented groups. Prior work addresses this issue by optimizing the worst-group performance. However, these approaches typically assume that the underlying data distributions for each group can be accurately estimated using the training data, a condition that is frequently violated in noisy, non-stationary, and evolving environments. In this work, we propose a novel framework that relies on Wasserstein-based distributionally robust optimization (DRO) to account for the distributional uncertainty within each group, while simultaneously preserving the objective of improving the worst-group performance. We develop a gradient descent-ascent algorithm to solve the proposed DRO problem and provide convergence results. Finally, we validate the effectiveness of our method on real-world data.

## 1 Introduction

Machine learning models are typically trained to minimize the average loss over training datasets, under the assumption that both training and testing samples are drawn independently from the same distribution. However, in real-world applications, this assumption is often violated. For instance, data can be generated from multiple heterogeneous environments, such as different hospitals, geographic regions, or demographic groups, and each of these environments can be associated with a distinct data distribution. In addition, even within a single environment, the data distribution may shift over time due to factors like temporal drift, changes in population demographics, or finite sampling bias. In these real-world applications, models trained without accounting for data heterogeneity or distribution shifts may show disparate performance across different subpopulations in the dataset – even if they achieve low average loss over the whole population Duchi et al. (2019) – or may even show average performance degradation when transferred from a training to a test set within the same environment. These shortcomings can be especially problematic in high-stakes domains like healthcare Seyyed-Kalantari et al. (2020) and finance Fuster et al. (2022); Khandani et al. (2010), where models should perform equally well across different population subgroups and maintain their performance in the presence of distribution shifts that can occur when they are deployed on environments that are different from those they were trained on.

A principled framework that has been widely used to introduce robustness to possible distribution shifts between training and test environments is Distributionally Robust Optimization (DRO). DRO employs a set of multiple plausible distributions that may describe future test environments, known as the ambiguity set, and formulates the robust learning problem as a min-sup problem that returns a model that minimizes the worst-case loss over this ambiguity set Goh and Sim (2010). In classical DRO, the ambiguity set is typically constructed as a ball around the data generating distribution of the training set, as shown in Figure 1a, with radius defined by different divergence measures such as $f$-divergence or Wasserstein distance Namkoong and Duchi (2016); Kuhn et al. (2019); Chen et al. (2018). While this formulation captures uncertainty around a single environment, it does not capture the effect of multi-source data that are common in practice.

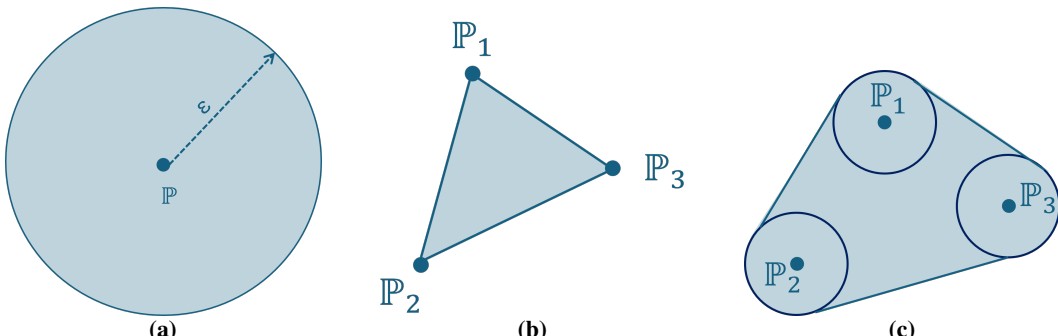

**Figure 1:** DRO ambiguity sets: (a) Ambiguity set in classical DRO that contains all distributions that are $\varepsilon$ away from the data generating distribution $\mathbb{P}$ that generated the training set; (b) Ambiguity set in classical Group DRO that contains all distributions that lie in the simplex formed by the data generating distributions $\{\mathbb{P}_1, \mathbb{P}_2, \mathbb{P}_3, \cdots\}$ of a collection of environments; and (c) Ambiguity set of our method that contains all distributions that are $\varepsilon_i$ away from the data generating distribution $\mathbb{P}_i$ of each environment $i = 1, 2, 3, \cdots$, as well as all mixtures of those.

To handle data generated from multiple environments with different data generating distributions, Group DRO (GDRO) methods have been proposed Sagawa et al. (2019); Oren et al. (2019). GDRO methods typically define an ambiguity set that contains all linear mixtures of those group distributions – Figure 1b – and formulate a min-max optimization problem that returns a model that performs well on the combination of environments with the worst expected loss. While this problem has been solved under perfect knowledge of the environments and their corresponding distributions using stochastic gradient update methods Sagawa et al. (2019); Zhang et al. (2023); Soma et al. (2022); Yu et al. (2024), there is limited work that considers uncertainty in the training data generating distributions. In this direction, the work in Ghosal and Li (2023) assumes that the training data labels in each group are uncertain and proposes a probabilistic group membership approach to address this challenge. However, addressing distribution shifts between the training and test sets in the local environments in a Group DRO setting is a problem that, to the best of our knowledge, still remains unexplored. *Our goal in this paper is to develop a unified framework that is robust both to heterogeneous data-generating environments and to within-group shifts.*

**Contributions.** We augment GDRO with explicit within-group uncertainty by introducing a grouped ambiguity set that includes all mixtures over groups and, for each group, all distributions within a small ball around its empirical distribution (Fig. 1c). This induces a nested min–max–sup objective in which adversarial within-group perturbations interact with group reweighting and parameter updates, making the extension nontrivial both computationally and analytically. We develop a tractable three-step gradient procedure that approximates the inner supremum via adversarial examples, updates group weights with exponentiated gradients, and performs stochastic parameter updates. Under standard assumptions we prove convergence to a stationary point. Experiments on real-world tabular and image datasets show consistent improvements over classical DRO and GDRO approaches.

## 2 PROBLEM SETUP: ACROSS-GROUP HETEROGENEITY AND WITHIN-GROUP SHIFT

We consider a training set $D_{train}$ consisting of data points sampled from $G$ environments with independent generating distributions $\{\mathbb{P}^1_{X,Y}, \mathbb{P}^2_{X,Y}, \cdots, \mathbb{P}^G_{X,Y}\}$, where $X \in \mathcal{X}$ and $Y \in \mathcal{Y}$ are random variables denoting covariates and outcomes, respectively. We assume that each data point in $D_{train}$ is of the form $\{x_i, y_i, g_i\}$ for $i = 1, \cdots, N$, where $x_i \in \mathcal{X}$ denotes the covariates, $y_i \in \mathcal{Y}$ denotes the outcomes, $g_i \in \{1, \cdots, G\}$ denotes the environment from which the point was sampled, and $N$ is the size of the dataset. We also assume that in the training set, a data-point is generated from an environment $g$ with probability $p_g$, such that $\sum_{g=1}^{G} p_g = 1$. Then, the data generating distribution $\mathbb{P}_{X,Y}$ associated with the training set $D_{train}$ can be defined as a mixture of the environmental distributions as $\mathbb{P}_{X,Y} = \sum_{g=1}^{G} p_g \cdot \mathbb{P}^g_{X,Y}$.

Using the data generated from $\mathbb{P}_{X,Y}$, typical machine learning techniques rely on Empirical Risk Minimization (ERM) to compute a parametric model $f_\theta : \mathcal{X} \to \mathcal{Y}$, where $\theta$ are the model parameters,

that minimizes the expected loss $\mathcal{L}$ over the training set, i.e.,

$$\min_\theta \mathbb{E}_{(x,y)\sim\mathbb{P}_{X,Y}}[\mathcal{L}(f_\theta; x, y)]. \tag{1}$$

## 2.1 DISTRIBUTION SHIFTS ACROSS GROUPS

When data are generated from multiple possibly heterogeneous environments, training a model on the distribution $\mathbb{P}_{X,Y}$ using equation 1, performs well on average but can lead to disparate performance across the individual environments Hong et al. (2023); Sagawa et al. (2019). In these situations, Group DRO can be used to improve the model performance across the different environments.

Group DRO Hu et al. (2018); Oren et al. (2019); Sagawa et al. (2019) takes into account the data generating distributions of the environments $\{\mathbb{P}^1_{X,Y}, \cdots, \mathbb{P}^G_{X,Y}\}$ and learns a model that performs best for the worst-case distribution among the groups. To this end, let $\mathcal{Q} := \left\{\sum_{g=1}^G q_g \mathbb{P}^g_{X,Y} : q \in \Delta_G\right\}$ denote an ambiguity set consisting of all linear combinations of the individual group distributions, where $q = [q_1, \cdots, q_G]$ denotes the vector of linear weights and $\Delta_G$ denotes the $G$-dimensional probability simplex. Then, the goal of Group DRO is to learn a model $f_\theta$ that optimizes the worst-case expected loss over the ambiguity set $\mathcal{Q}$, i.e.,

$$\min_{\theta\in\Theta} \sup_{Q\in\mathcal{Q}} \mathbb{E}_{(x,y)\sim Q}[\mathcal{L}(f_\theta; x, y)]. \tag{2}$$

Since each distribution $Q \in \mathcal{Q}$ is a weighted combination of the local group distributions $\mathbb{P}^g_{X,Y}$, Group DRO effectively searches over the space of possible weighted combinations of the local group distributions to emphasize those with higher loss. Therefore, instead of biasing the model toward environments that dominate the training set, as in the case of ERM, Group DRO encourages the model to improve performance on the most challenging or under-performing environments.

Finally, since each distribution $Q \in \mathcal{Q}$ is a weighted combination of the local group distributions $\mathbb{P}^g_{X,Y}$, the expected loss over $Q$ can be written as

$\mathbb{E}_{(x,y)\sim Q}[\mathcal{L}(f_\theta; x, y)] = \sum_{g=1}^G q_g \mathbb{E}_{(x,y)\sim\mathbb{P}_g}[\mathcal{L}(f_\theta; x, y)] = \sum_{g=1}^G q_g \mathcal{L}_g(f_\theta),$

where $\mathcal{L}_g(f_\theta) := \mathbb{E}_{(x,y)\sim\mathbb{P}_g}[\mathcal{L}(f_\theta; x, y)]$ denotes the expected group-level loss. Substituting this expectation into equation 2, we obtain an equivalent min-max formulation of the Group DRO problem as

$$\min_{\theta\in\Theta} \max_{q\in\Delta_G} \sum_{g=1}^G q_g \mathcal{L}_g(f_\theta). \tag{3}$$

## 2.2 DISTRIBUTION SHIFTS ACROSS AND WITHIN GROUPS

Different to existing literature, in this paper we assume that the data generating distribution $\mathbb{P}^g_{X,Y}$ in each environment is unknown. Even though the empirical distribution $\hat{\mathbb{P}}^g_{X,Y}$ of each group $g \in \{1, \cdots, G\}$ can be estimated from training data and used to approximate the true data generating distribution $\mathbb{P}^g_{X,Y}$, this approximation can contain errors due to finite-sampling bias or possible distribution shifts.

To address uncertainty in the data generating distributions in the local environments, we extend the group DRO framework discussed in Section 2.1 by combining it with a local DRO objective at each local environment. The goal is to learn models that are robust to both changes in the mixture of the local environments as well as to distribution shifts within each of the local environments. Specifically, given a distance metric $D$ between distributions, let $\mathcal{P}_g = \{\mathbb{P} : D(\mathbb{P}, \hat{\mathbb{P}}^g_{X,Y}) \leq \epsilon_g\}$ denote an ambiguity set containing all possible data generating distributions for group $g \in \{1, \cdots, G\}$, that is a ball of radius $\epsilon_g > 0$ around the empirical distribution $\hat{\mathbb{P}}^g_{X,Y}$. As a distance metric between two distributions we use the 1-Wasserstein metric defined below.

**Definition 1** (1-Wasserstein Distance Villani (2009)). *Let $\mathbb{P}$ and $\mathbb{P}'$ be two probability distributions on a Polish space $\Xi \subseteq \mathbb{R}^d$ with finite second moments. Let $\Gamma(\mathbb{P}, \mathbb{P}')$ denote the set of all couplings (i.e., joint distributions) with marginals $\mathbb{P}$ and $\mathbb{P}'$ and let $c : \Xi \times \Xi \to [0, \infty)$ denote the transportation cost. Then, the 1-Wasserstein distance between $\mathbb{P}$ and $\mathbb{P}'$ is defined as* $W_1(\mathbb{P}, \mathbb{P}') := \inf_{\gamma\in\Gamma(\mathbb{P},\mathbb{P}')} \int_{\Xi\times\Xi} c(x, y) \, d\gamma(x, y).$

---

**Algorithm 1** Gradient Ascent for Worst-case Perturbations

---

**Require:** $(x, y)$, $\eta_z$, $T_{rob}$, cost function $c : (\mathcal{X} \times \mathcal{Y}) \times (\mathcal{X} \times \mathcal{Y}) \to \mathbb{R}_+$ and model parameters $f_\theta$
1: Initialize $z^0$ with $(x, y)$
2: **for** $t = 1$ to $(T_{rob})$ **do**
3:     $\phi(f_\theta; (x, y), z^{t-1}) = \mathcal{L}(f_\theta; z^{t-1}) - \gamma c((x, y), z^{t-1})$
4:     $z^t \leftarrow z^{t-1} + \eta_z \nabla_z \phi(f_\theta; (x, y), z^{t-1})$
5: **end for**
6: Return $z^{T_{rob}}$

---

Given the uncertainty set $\mathcal{P}_g$ for each environment $g \in \{1, \cdots, G\}$, we can define the robust group-level loss as

$$\mathcal{L}_g^{ROB}(f_\theta) = \sup_{\mathbb{P}_{X,Y}^g \in \mathcal{P}_g} \mathbb{E}_{(x,y) \sim \mathbb{P}_{X,Y}^g}[\mathcal{L}(f_\theta; x, y)]. \tag{4}$$

Substituting the robust group-level loss into equation 3, we obtain the proposed group DRO under group-level distributional uncertainty problem with the objective to learn a model $f_\theta$ that is robust to both the worst-case mixture of environments and to distribution shifts within environments, i.e.,

$$\min_{\theta \in \Theta} \max_{q \in \Delta_G} \sum_{g=1}^{G} q_g \mathcal{L}_g^{ROB}(f_\theta). \tag{5}$$

## 3 METHODOLOGY

In this section, we introduce a gradient method to solve the min-max Group DRO with group-level distributional uncertainty problem equation 5. In particular, we design an iterative algorithm that at each iteration first computes the robust loss $\mathcal{L}_g^{ROB}$ for each environment $g \in \{1, \cdots, G\}$, and then performs gradient descent mirror ascent steps to update the parameters of the min-max Group DRO problem. In the following subsections we analyze these two parts of our algorithm.

### 3.1 LAGRANGIAN RELAXATION OF THE ROBUST GROUP-LEVEL LOSS

Directly computing the robust group-level loss $\mathcal{L}_g^{ROB}$ for every group $g \in \{1, \cdots, G\}$ is generally intractable, since it requires computing the supremum over an infinite ambiguity set of distributions $\mathcal{P}_g$. When the model $f_\theta$ is convex with respect to the model parameters $\theta$, e.g., linear Chen and Paschalidis (2018) or a logistic regression Shafieezadeh Abadeh et al. (2015), the dual formulation of the robust loss in equation 4 can be cast as a convex optimization problem that can be efficiently solved using existing solvers.

However, when the model $f_\theta$ is not convex with respect to the model parameters $\theta$ as, e.g., in the case of Neural Networks, to the best of our knowledge, tractable dual formulations of the robust loss equation 4 do not exist. In this case, to compute the robust loss we instead resort to its Lagrangian relaxation

$$\mathcal{L}_{g,\gamma}^{ROB}(f_\theta) = \sup_{\mathbb{P}_{X,Y}^g \in \mathcal{P}_g} \left[ \mathbb{E}_{(x,y) \sim \mathbb{P}_{X,Y}^g}[\mathcal{L}(f_\theta; x, y)] - \gamma W_1(\mathbb{P}_g, \hat{\mathbb{P}}_g) \right], \tag{6}$$

where $\gamma \geq 0$ is a fixed penalty parameter. The Lagrangian relaxation of the robust loss in equation 6 can be efficiently computed as shown in the following result.

**Proposition 3.1.** *(Proposition 1 in Sinha et al. (2017)) Let $\mathcal{L} : \Theta \times (\mathcal{X}, \mathcal{Y}) \to \mathbb{R}$ be a loss function and $c : (\mathcal{X}, \mathcal{Y}) \times (\mathcal{X}, \mathcal{Y}) \to \mathbb{R}_+$ be a continuous transportation cost function. Then, for any distribution $\hat{\mathbb{P}}_{X,Y}^g$, $\gamma \geq 0$, and any uncertainty set $\mathcal{P} = \{\mathbb{P} : W_1(\mathbb{P}, \hat{\mathbb{P}}_{X,Y}^g) \leq \epsilon\}$ we have*

$$\mathcal{L}_{g,\gamma}^{ROB}(f_\theta) = \mathbb{E}_{(x,y) \sim \hat{\mathbb{P}}_{X,Y}^g} \left[ \sup_{(x',y') \in \mathcal{X} \times \mathcal{Y}} \phi(f_\theta; (x, y), (x', y')) \right], \tag{7}$$

*where $\phi(f_\theta; (x, y), (x', y')) = \mathcal{L}(f_\theta; x', y') - \gamma c((x, y), (x', y'))$ is a penalized loss.*

Proposition 3.1 allows to compute the robust loss $\mathcal{L}_{g,\gamma}^{ROB}$ of each group $g \in \{1, \cdots, G\}$ by exhaustively searching over the support $\mathcal{X} \times \mathcal{Y}$ for points $(x', y')$ that maximize the penalized loss $\phi(f_\theta; (x, y), (x', y'))$. However, when the support $\mathcal{X} \times \mathcal{Y}$ is large or infinite – e.g., for continuous

---

**Algorithm 2** Group DRO with distributional uncertainty per group

---

**Require:** Training set $D_{train} = \{x_i, y_i, g_i\}_{i=1}^N$, model $f_\theta$, number of iterations $T$, learning rates $\{\eta_\theta, \eta_q\}$, cost parameter $\gamma$, cost function $c : (\mathcal{X} \times \mathcal{Y}) \times (\mathcal{X} \times \mathcal{Y}) \to \mathbb{R}_+$
1: Initialize $\theta_0 \in \Theta$ randomly and $q_g^0 = {N_g}/{N} \ \forall g \in \{1, \cdots, G\}$
2: **for** $t = 1$ to $T$ **do**
3:     $B_t = []$
4:     **for** $(x_i, y_i, g_i) \in D_{train}$ **do**
5:         Use Algorithm 1 to obtain $z_i = arg\max_{z \in (\mathcal{X} \times \mathcal{Y})} \phi(f_{\theta_{t-1}}; (x_i, y_i), z)$
6:     **end for**
7:     **for** $g \in \{1, \cdots, G\}$ **do**
8:         $\mathcal{L}_{g,\gamma}^{ROB}(f_{\theta_{t-1}}) = \frac{1}{N_g} \sum_{i=1}^{N_g} \phi(f_{\theta_{t-1}}; (x_i, y_i), z_i)$
9:         $\nabla\mathcal{L}_{g,\gamma}^{ROB}(f_{\theta_{t-1}}) = \frac{1}{N_g} \sum_{i=1}^{N_g} \frac{\partial \phi(f_{\theta_{t-1}}; (x_i, y_i), z_i)}{\partial \theta}$
10:     **end for**
11:     $m_g^t \leftarrow q_g^{t-1} \exp\left\{\eta_q \mathcal{L}_{g,\gamma}^{ROB}(f_{\theta_{t-1}})\right\} \forall g \in \{1, \cdots, G\}$
12:     $q_g^t \leftarrow \frac{m_g^t}{\sum_{g=1}^G m_g^t} \forall g \in \{1, \cdots, G\}$
13:     $\theta_t \leftarrow \theta_{t-1} - \eta_\theta \sum_{g=1}^G q_g^t \nabla\mathcal{L}_{g,\gamma}^{ROB}(f_{\theta_{t-1}})$
14: **end for**

---

variables – searching over the support can be computationally intractable. In this case, similar to the work in Sinha et al. (2017), we propose a gradient ascent algorithm to instead approximate the robust loss $\mathcal{L}_{g,\gamma}^{ROB}$. The proposed algorithm is summarized in Algorithm 1. Specifically, for every sample $(x, y) \in D_{train}$ in the training set, Algorithm 1 iteratively updates the point $z = (x', y') \in \mathcal{X} \times \mathcal{Y}$ to maximize the penalized loss $\phi(f_\theta; (x, y), (x', y'))$.

### 3.2 SOLUTION OF THE GROUP DRO WITH GROUP-LEVEL DISTRIBUTIONAL UNCERTAINTY PROBLEM

In this section we present an algorithm to solve the min-max group DRO with group-level distributional uncertainty problem equation 5. The proposed algorithm is summarized in Algorithm 2. Specifically, Algorithm 2 is an iterative method that consists of the following steps. First, for each point in the training dataset $(x_i, y_i, g_i) \in D_{train}$, Algorithm 1 is used to compute the point $z_i$ that, given the current model parameters $\theta_{t-1}$, maximizes the penalized loss $\phi(f_{\theta_{t-1}}; (x_i, y_i), z_i)$, as shown in lines 4-6. Then, using equation 7 and the point $z_i$, Algorithm 2 computes the approximate expected robust loss $\mathcal{L}_{g,\gamma}^{ROB}(f_{\theta_{t-1}})$ and its gradient with respect to $\theta$, $\nabla\mathcal{L}_{g,\gamma}^{ROB}(f_{\theta_{t-1}})$, for the current model $\theta_{t-1}$ and each environment $g \in \{1, \cdots, G\}$, as shown in lines 7-10. Next, using the robust losses computed in line 8, the weights $q_g^t$ for each environment $g \in \{1, \cdots G\}$ are updated by a mirror ascent step, as shown in lines 11-12. Finally, given the updated weights $q_g^t$ and the robust gradients for each environment, Algorithm 2 updates the model parameters $\theta_t$ by a gradient descent step, as shown in line 13. The Convergence Analysis of the proposed Algorithm is provided in the Appendix.

## 4 NUMERICAL EXPERIMENTS

In this section, we evaluate the performance of our proposed method against several baseline approaches on real-world datasets. Our framework is designed to address both heterogeneity across different subpopulations in the data and possible distribution shifts within individual subpopulations between the training and test sets. Through our experiments, we aim to demonstrate that failing to account for either of these two objectives can lead to sub-optimal performance when both are existent in the data. To this end, we compare against three baseline methods: (1) Empirical Risk Minimization (ERM) that trains a model to minimize the average loss over the entire training set without accounting for group identities or distribution uncertainties, (2) Distributionally Robust Optimization (DRO) as proposed in Sinha et al. (2017) that accounts for distributional shifts between the training and test sets but ignores group structure, and (3) Group DRO (GDRO) as proposed in Sagawa et al. (2019) that explicitly addresses group-wise performance disparities but assumes full knowledge of each group's data generating distribution without modeling within-group distributional uncertainty.

We evaluate all methods on test sets that have not been observed during training. To assess the effectiveness of each method, we focus on their accuracy. For each group $g \in \{1, \cdots, G\}$ with a test dataset $D_{test}^g = \{(x_i^g, y_i^g)\}_{i=1}^{N_g}$, we define the accuracy of a model $f_\theta$ as $Acc_g = \sum_{i=1}^{N_g} \mathbb{1}\{y_i^g == f_\theta(x_i^g)\}/N_g 100\%$. Given the accuracy for each group, we report three evaluation metrics:

(i) the average accuracy across all groups **Average Accuracy** $:= \sum_{g=1}^{G} Acc_g/G$,

(ii) the range of accuracies among groups

**Range of Accuracy** $:= \max_{g \in \{1, \cdots, G\}} Acc_g - \min_{g \in \{1, \cdots, G\}} Acc_g$, and

(iii) the worst-case accuracy observed across groups

**Worst-Case Accuracy** $:= \min_{g \in \{1, \cdots, G\}} Acc_g$

Consistent with the evaluation protocol in Sagawa et al. (2019), we report both the average accuracy and the worst-case group accuracy to assess overall model performance and its ability to generalize to the most under-performing group. In addition to these metrics, we include the accuracy range across groups to quantify performance disparities, thereby providing a complementary measure of fairness and consistency in group-level outcomes. The higher the average and worst-case accuracy and the lower the range of accuracy, the better a model's performance.

## 4.1 Tabular Classification under Distributional Shift

We evaluate how the models perform on two tabular datasets. The Adult Income dataset Becker and Kohavi (1996) and a publicly available brain stroke dataset Hassan (2023).

**Adult** is a widely used benchmark of 47,621 individuals with 15 demographic and occupational features (e.g., age, race, gender, education level, marital status, occupation) and a binary label indicating whether annual income exceeds $50,000. It is frequently used in robustness studies because of pronounced group heterogeneity. Prior work (e.g., Soma et al. (2022); Zhang et al. (2023)) typically defines groups using sensitive attributes such as race and gender. Following Sagawa et al. (2019), we adopt six intersectional groups based on race $\{\text{White}, \text{Black}, \text{Other}\}$ crossed with the income label $\{\leq 50K, > 50K\}$.

The **Stroke** dataset contains medical and demographic information for 5,110 patients. In particular it includes 11 features per patient, such as age, gender, and average glucose level, along with a binary outcome indicating whether the patient experienced a stroke. Notably, the dataset does not provide a temporal relationship between when measurements were recorded for a patient and when the stroke event occurred. Additionally, it is highly imbalanced, with only 249 patients having experienced a stroke, and the remaining 4,861 not, which poses challenges for training fair and robust classifiers. In this case, we adopt a group definition based on age $\{\leq 60, > 60\}$ and stroke outcome $\{\text{positive}, \text{negative}\}$.

For both cases, we run ten independent trials with fixed seeds and show the mean and variance of the performance across these seeds. Our goal is twofold: first, to compare our method against baselines for robustness across the above groups; second, to assess robustness under a train–test distribution shift. To induce a controlled covariate shift, we construct training splits with a uniform marginal over the attribute of education for the Adult dataset and smoking for the stroke dataset. We evaluate on test splits that follow the original attribute distributions. More details about the experiments can be found in the Appendix.

For the robustness parameter $\gamma$, we sweep $\gamma \in \{10^{-4}, 10^{-3}, 10^{-2}, 10^{-1}, 0.5, 1, 3, 5, 6, 7, 8\}$ to select the best value via fine-tuning and to study sensitivity. The remaining hyperparameters in Algorithm 1 are $\eta_\theta = 0.1$, $\eta_q = 0.1$, $\eta_z = 0.05$, $T_{\text{rob}} = 100$, and $T = 200$.

Numerical results in Table 1 show consistent trends across both tasks. **Adult**: ERM and standard DRO offer limited subgroup robustness (avg $\approx 0.55/0.52$, worst-group $\approx 0.08/0.03$, range $\approx 0.92/0.97$), while Group DRO markedly improves subgroup parity (worst-group $\approx 0.56$, range $\approx 0.26$). Our method achieves the best trade-off (avg $0.715 \pm 0.011$, worst-group $0.613 \pm 0.061$, range $0.193 \pm 0.093$). **Stroke**: ERM and DRO again underperform on worst-group performance (avg $\approx 0.50$, worst-group $\approx 0.00$, range $\approx 1.00$); Group DRO improves both worst-group and disparity (avg $0.630 \pm 0.032$, worst-group $0.493 \pm 0.074$, range $0.270 \pm 0.079$); and our method performs best overall (avg $0.666 \pm 0.019$, worst-group $0.593 \pm 0.032$, range $0.202 \pm 0.094$). Across both datasets, modeling within-group uncertainty on top of group reweighting yields higher worst-group performance and smaller disparities without sacrificing average accuracy.

**Table 1:** Evaluation results of the four methods for the Income and the Stroke prediction Tasks. Best is strong blue, runner-up light blue, worst light red.

| | **Income Prediction** | | | |
| --- | --- | --- | --- | --- |
| | ERM | DRO ($\gamma$=9) | Group DRO | Ours ($\gamma$=$10^{-4}$) |
| Average Accuracy | $0.5471\pm0.0100$ | $0.5165\pm0.0037$ | $0.6953\pm0.0269$ | $0.7148\pm0.0105$ |
| Worst-Group Accuracy | $0.0815\pm0.0220$ | $0.0287\pm0.0073$ | $0.5607\pm0.0388$ | $0.6126\pm0.0605$ |
| Accuracy Range | $0.9154\pm0.0241$ | $0.9709\pm0.0077$ | $0.2571\pm0.0789$ | $0.1934\pm0.0927$ |
| | **Stroke Prediction** | | | |
| | ERM | DRO ($\gamma$=9) | Group DRO | Ours ($\gamma$=$10^{-2}$) |
| Average Accuracy | $0.5\pm0.0002$ | $0.5\pm0.0$ | $0.6295\pm0.0321$ | $0.6664\pm0.0194$ |
| Worst-Group Accuracy | $0.0\pm0.0$ | $0.0\pm0.0$ | $0.4926\pm0.0735$ | $0.5927\pm0.0316$ |
| Accuracy Range | $1.0\pm0.0$ | $1.0\pm0.0$ | $0.2697\pm0.0789$ | $0.2017\pm0.0935$ |

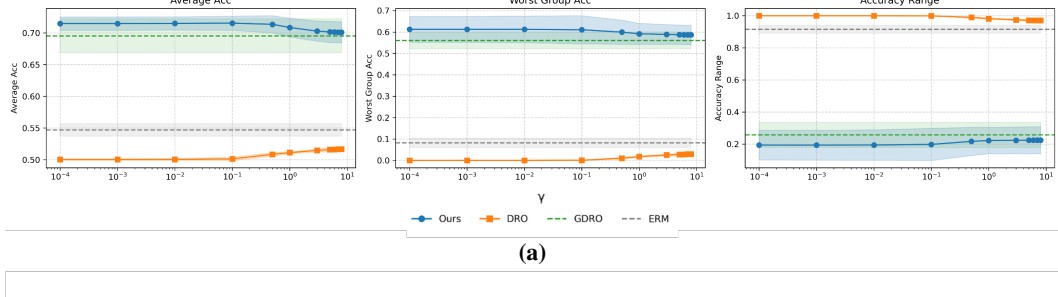

(a)

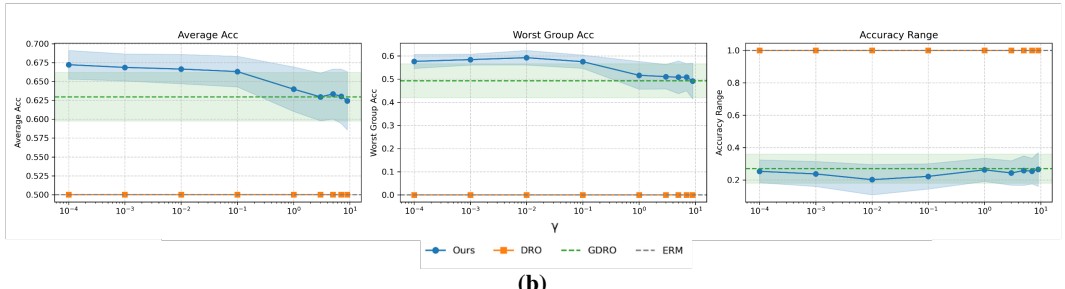

(b)

**Figure 2:** Performance of ERM, DRO, Group DRO, and our method under distribution shift as a function of $\gamma$: (a) `Adult` dataset with shift on *education*; (b) `Stroke` dataset with shift on *smoking*.

Figure 2 illustrates how performance evolves as $\gamma$ is varied. Across both datasets, our method remains consistently stable, maintaining high average accuracy, strong worst-group accuracy, and low disparity. By contrast, standard DRO does not meaningfully improve across the sweep, with worst-group accuracies near 0 and accuracy ranges close to 1. Together, these results confirm that our approach not only outperforms all baselines numerically, but also remains robust and reliable across hyperparameter choices.

## 4.2 TABULAR CLASSIFICATION WITH EVALUATION ON MULTIPLE ENVIRONMENTAL SHIFTS

In Section 4.1, we considered a single controlled covariate shift between training and test distributions based on either the `education` or the `smoking` attribute. We now extend this analysis by evaluating model performance across *multiple test environments* in order to more thoroughly assess robustness.

The models are trained on the same training datasets as in Section 4.1, where the marginal distribution of `education` is uniform. To study robustness under shifting environments, we split the values of the attributes `education` (for Adult) and `smoking` (for Stroke) into two groups and vary their relative proportions to create families of test distributions. More details on how the test environments were constructed can be found in the Appendix.

This experimental design allows us to probe the extent to which each method is robust to unseen environmental changes, going beyond a single fixed train–test split. In practice, such variations are

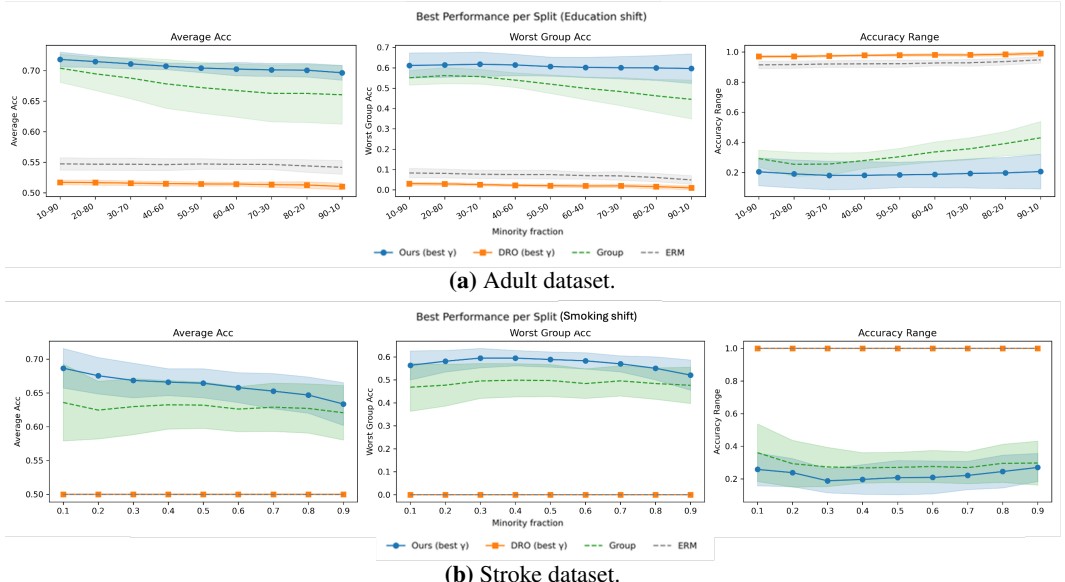

**(a)** Adult dataset.

**(b)** Stroke dataset.

**Figure 3:** Best performance across group splits for all methods. Models are trained with uniform training distributions (`education` for Adult, `smoking` for Stroke). Test environments are constructed with varying proportions of the two groups (from 90–10 to 10–90). Plotted are the average accuracy, worst-group accuracy, and accuracy range of each method at their best-performing $\gamma$ (if applicable).

common in real-world deployment, where underlying demographics and education levels can vary substantially across regions or over time.

We report the best performance of each method across the constructed environments in Figure 3. The trends are consistent across both datasets. ERM and standard DRO struggle under severe imbalances: worst-group accuracies approach zero and disparities remain large (accuracy ranges near 1). Group DRO mitigates these issues by raising worst-group accuracy and narrowing gaps, though variability across environments remains. In contrast, our method achieves the highest and most stable performance overall, indicating both improved accuracy and robustness to a wide spectrum of distributional shifts.

### 4.3 IMAGE CLASSIFICATION UNDER DISTRIBUTIONAL SHIFT

We next evaluate our method on a widely used vision robustness benchmark Lee et al. (2025).

**Colored MNIST** introduces a controlled spurious attribute for digit classification. We consider the binary task of predicting whether a digit is $< 5$ or $\geq 5$. In the *training* set, color is strongly aligned with the label (red for $< 5$, green for $\geq 5$ with high probability), making color highly predictive; groups are defined by the joint of label ($< 5/\geq 5$) and color (red/green). To induce a single train–test distribution shift, we (i) change class composition—overrepresenting digits $\{0, 1, 2, 5, 6, 7\}$ in training and $\{3, 4, 8, 9\}$ in testing—and (ii) neutralize the color–label correlation at test (approximately 50–50 red/green regardless of label), reducing reliance on the spurious cue and stressing robustness.

Similar to the previous cases we run ten independent trials with fixed seeds. In each of them we sweep $\gamma \in \{10^{-4}, 10^{-3}, 10^{-2}, 10^{-1}, 0.5, 1, 3, 5, 6, 7, 8\}$ The remaining hyperparameters in Algorithm 1 are $\eta_\theta = 0.1$, $\eta_q = 0.1$, $\eta_z = 0.05$, $T_{\text{rob}} = 50$, and $T = 30$.

Figure 4 summarizes the results in this setting and shows that our method achieves the best balance between average performance and subgroup robustness. While average accuracy is competitive with DRO and ERM, our method substantially improves worst-group accuracy, reaching $\approx 0.95$ compared to $\approx 0.90$ for DRO and $\approx 0.91$ for ERM. At the same time, it achieves the smallest accuracy range across groups ($\approx 0.02$–$0.04$), whereas other methods maintain much larger disparities. Overall, our approach not only preserves high average accuracy but also ensures equitable group performance, consistent with the trends observed on the tabular datasets.

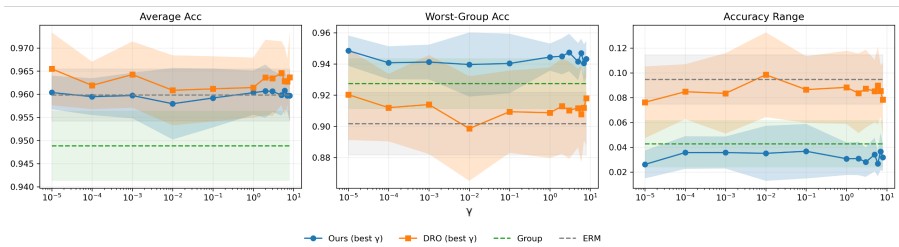

**Figure 4:** Performance of ERM, DRO, Group DRO, and our method on the `Colored MNIST` dataset under distribution shift as a function of $\gamma$.

**Additional multimodal results (CheXpert).** For completeness, we also evaluate our approach on a multimodal pneumonia prediction task using CheXpert Irvin et al. (2019); full dataset details, model, and results are deferred to Appendix C. The conclusions mirror our tabular and MNIST studies: our method improves worst-group accuracy and narrows disparities while maintaining competitive average accuracy.

## 5 DISCUSSION

The results across Sections 4.1–4.2 highlight the importance of explicitly addressing both cross-group heterogeneity and distributional uncertainty in robust learning. In all tasks, Empirical Risk Minimization (ERM) is shown to be inadequate: it attains moderate average accuracy but underperforms on minority subgroups, with very low worst-group accuracy and large disparities. This behavior reflects ERM's tendency to overfit majority-dominated patterns, yielding models that are brittle under imbalance or shift.

Standard DRO, while aiming for robustness, inherits similar limitations. Because a single ambiguity set is centered on the overall empirical distribution, the objective is dominated by well-sampled majority groups and offers limited protection for rare or underrepresented subpopulations. In some settings it performs on par with—or below—ERM on worst-group metrics, underscoring that classical distributional robustness without group structure cannot adequately safeguard vulnerable populations under domain shift.

Group DRO (GDRO) directly targets the worst-performing group and typically improves worst-group accuracy while narrowing disparities. However, these gains can come at a cost to mean accuracy when small or noisy groups are overweighted. Moreover, in the *tabular* multi-environment experiments (Section 4.2), GDRO's improvements are not uniformly sustained as covariate distributions vary more dramatically across test environments, reflecting sensitivity to how subgroup distributions shift.

Our method achieves the strongest and most stable overall performance across scenarios. In single-environment shifts, it improves worst-group accuracy and reduces disparities without sacrificing mean accuracy. In the *tabular* multi-environment analyses, it maintains high accuracy, strong worst-group performance, and low disparity across a wide range of shifts. For the *image* task, where we vary only the inner robustness parameter $\gamma$ (rather than constructing multiple environments), the method remains stable across broad $\gamma$ sweeps, again sustaining worst-group performance alongside high average accuracy.

A practical consideration is the choice of the robustness parameter $\gamma$. Unlike a Wasserstein radius $\epsilon$ with a geometric interpretation, $\gamma$ appears as a Lagrange penalty and lacks direct physical meaning. Nevertheless, performance is remarkably stable across orders of magnitude (e.g., $\gamma \in [10^{-4}, 10^1]$), suggesting that fine tuning is unnecessary in practice. Conceptually, very small $\gamma$ behaves closer to standard DRO, very large $\gamma$ approaches GDRO, and intermediate values consistently deliver favorable trade-offs between fairness and robustness without harming average accuracy.

Taken together, these findings underscore two messages. First, robust learning must go beyond ERM and standard DRO to explicitly account for both group structure and intra-group uncertainty. Second, our method offers a flexible and effective way to do so: it interpolates between DRO and GDRO, attains state-of-the-art trade-offs in worst-group and mean accuracy, and shows particular strength when distributional shifts are most pronounced.

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
