# Appendix

## A CONVERGENCE ANALYSIS

### A.1 PRELIMINARIES

We first recall a few standard notions for smooth and weakly-convex functions on $\mathbb{R}^d$.

**Definition 2** (Lipschitz continuity). *A function $f : \mathbb{R}^d \to \mathbb{R}$ is $L$-Lipschitz if for all $x, x' \in \mathbb{R}^d$,*

$$\|f(x) - f(x')\| \leq L \|x - x'\|.$$

**Definition 3** (Smoothness). *A differentiable function $f : \mathbb{R}^d \to \mathbb{R}$ is $\ell$-smooth if for all $x, x' \in \mathbb{R}^d$,*

$$\|\nabla f(x) - \nabla f(x')\| \leq \ell \|x - x'\|.$$

Consider the problem

$$\min_x \max_y f(x, y).$$

Given this min-max problem we define

$$\Phi(x) = \max_{y \in Y} f(x, y),$$

where $f(x, \cdot)$ is concave on a convex, bounded set $Y$. Even though $\Phi$ may be nonconvex, one can still seek *stationary points* of $\Phi$ as a proxy for global minimizers.

**Definition 4** (Stationarity — differentiable case). *A point $x \in \mathbb{R}^d$ is an $\varepsilon$-stationary point of a differentiable $\Phi$ if*

$$\|\nabla \Phi(x)\| \leq \varepsilon.$$

*When $\varepsilon = 0$, $x$ is a true stationary point.*

If $\Phi$ is not differentiable (e.g. in the general nonconvex-concave setting), we weaken this via *weak convexity*.

**Definition 5** (Weak convexity). *A function $\Phi : \mathbb{R}^d \to \mathbb{R}$ is $\ell$-weakly convex if*

$$x \mapsto \Phi(x) + \tfrac{\ell}{2}\|x\|^2$$

*is convex.*

In particular, one can define the *Moreau envelope* of $\Phi$, which both smooths and regularizes it.

**Definition 6** (Moreau envelope). *For $\lambda > 0$, the* Moreau envelope *of $\Phi$ is*

$$\Phi_\lambda(x) = \min_{w \in \mathbb{R}^d} \left\{ \Phi(w) + \tfrac{1}{2\lambda}\|w - x\|^2 \right\}.$$

**Lemma A.1** (Smoothness of the Moreau envelope). *If $f$ is $\ell$-smooth and $Y$ is bounded, then the envelope $\Phi_{1/(2\ell)}$ of $\Phi(x) = \max_{y \in Y} f(x, y)$ is differentiable, $\ell$-smooth, and $\ell$-strongly convex.*

This allows an alternative stationarity measure:

**Definition 7** (Stationarity via Moreau envelope). *A point $x$ is $\varepsilon$-stationary for an $\ell$-weakly convex $\Phi$ if*

$$\left\|\nabla \Phi_{1/(2\ell)}(x)\right\| \leq \varepsilon.$$

**Lemma A.2** (Proximity to ordinary subgradients). *If $x$ satisfies $\|\nabla \Phi_{1/(2\ell)}(x)\| \leq \varepsilon$, then there exists $\hat{x}$ such that*

$$\min_{\xi \in \partial \Phi(\hat{x})} \|\xi\| \leq \varepsilon \quad and \quad \|x - \hat{x}\| \leq \tfrac{\varepsilon}{2\ell}.$$

Finally, since our algorithm performs a mirror-ascent update on the dual variable $q \in \Delta_G$, we require some standard facts about the associated Bregman divergence on the probability simplex. Concretely, let

$$\varphi \colon \mathbb{R}^G \to \mathbb{R} \quad \text{be a strictly convex } C^1 \text{ generator.}$$

Then for any $p_1, p_2 \in \Delta^{n-1}$ the *Bregman divergence* is defined by

$$D_\varphi(p_1 \parallel p_2) = \varphi(p_1) - \varphi(p_2) - \langle \nabla\varphi(p_2), p_1 - q \rangle.$$

**Definition 8** (Legendre generator). *A function $\varphi$ is called a* Legendre generator *on the simplex if*

1. *$\varphi$ is strictly convex and continuously differentiable on the open simplex $\{x > 0, \sum_i x_i = 1\}$,*

2. *its gradient $\nabla\varphi$ extends continuously to the closed simplex $\Delta^{n-1}$,*

3. *and $\varphi$ attains its global minimum at the uniform distribution $u = (1/n, \ldots, 1/n)$.*

**Example A.1** (Negative-entropy / KL generator). *When*

$$\varphi(x) = \sum_{i=1}^{n} x_i \ln x_i,$$

*one obtains the Kullback–Leibler divergence,*

$$D_{\mathrm{KL}}(p_1 \| p_2) = \sum_{i=1}^{n} p_{1,i} \ln \frac{p_{1,i}}{p_{2,i}}.$$

*Since we initialize and maintain all iterates in the interior of $\Delta_G$, this divergence remains finite throughout our mirror-ascent steps.*

**Property A.1** (Nonnegativity & convexity). *For any Legendre generator $\varphi$, one has*

$$D_\varphi(p\|q) \geq 0, \quad D_\varphi(p\|q) = 0 \iff p = q,$$

*and $D_\varphi(\,\cdot\,\|q)$ is convex in its first argument.*

**Lemma A.3** (Boundedness on the simplex). *If both $\varphi$ and $\nabla\varphi$ are bounded on the closed simplex, then*

$$\max_{p_1, p_2 \in \Delta^{n-1}} D_\varphi(p_1 \| p_2) < \infty.$$

**Lemma A.4** (KL-bound under interior iterates). *Suppose during mirror-ascent every dual iterate $q_t \in \Delta_G$ satisfies*

$$q_{t,i} \geq \delta \quad \forall\, i = 1, \ldots, G, \ t = 0, 1, \ldots,$$

*for some $\delta > 0$. Then for any two such iterates $p_1, p_2 \in \Delta_G$,*

$$D_{\mathrm{KL}}(p_1 \| p_2) \leq D.$$

*Where $D = ln\frac{1}{\delta}$.*

*Proof.* Since $p_{2,i} \geq \delta$ for all $i$, we have

$$\ln \frac{p_{1,i}}{p_{2,i}} \leq \ln \frac{p_{1,i}}{\delta} = \ln \frac{1}{\delta} + \ln p_{1,i},$$

and because $\sum_i p_{1,i} = 1$ and $\sum_i p_{1,i} \ln p_{1,i} \leq 0$,

$$D_{\mathrm{KL}}(p_1 \| p_2) = \sum_i p_{1,i} \ln \frac{p_{1,i}}{p_{2,i}} \leq \sum_i p_{1,i} \ln \frac{1}{\delta} + \sum_i p_{1,i} \ln p_{1,i} \leq \ln \frac{1}{\delta}.$$

$\square$

## A.2 PROPERTIES OF THE ROBUST LOSS

**Assumption A.1.** *The cost function $c : (\mathcal{X} \times \mathcal{Y} \times \mathcal{X} \times \mathcal{Y}) \to \mathbb{R}_+$ is continuous. For each $x \in \mathcal{X}$ and $y \in \mathcal{Y}$, $c(\cdot, (x, y))$ is 1-strongly convex with respect to the norm $||\cdot||$.*

**Assumption A.2** (Lipschitz Loss Function). *Consider the loss function $\mathcal{L}$. Then for every function $f$ with model parameters $\theta \in \Theta$ and for every $(x, y) \in (\mathcal{X}, \mathcal{Y})$ we assume that $\mathcal{L}$ is K-Lipschitz with respect to $\theta$*

**Assumption A.3** (Lipschitz smoothness of the loss). *Consider the loss function $\mathcal{L}$. Then for every function $f$ with model parameters $\theta \in \Theta$ and for every $(x, y) \in (\mathcal{X}, \mathcal{Y})$ we assume*

$$\left\| \nabla_\theta \, \mathcal{L}(f_\theta; x, y) \, - \, \nabla_\theta \, \mathcal{L}(f_{\theta'}; x, y) \right\| \leq L_{\theta\theta} \, \|\theta - \theta'\|,$$
$$\left\| \nabla_{x,y} \, \mathcal{L}(f_\theta; x, y) \, - \, \nabla_{x,y} \, \mathcal{L}(f_\theta; x', y') \right\| \leq L_{zz} \, \|(x, y) - (x', y')\|,$$
$$\left\| \nabla_\theta \, \mathcal{L}(f_\theta; x, y) \, - \, \nabla_\theta \, \mathcal{L}(f_\theta; x', y') \right\| \leq L_{\theta z} \, \|(x, y) - (x', y')\|,$$
$$\left\| \nabla_{x,y} \, \mathcal{L}(f_\theta; x, y) \, - \, \nabla_{x,y} \, \mathcal{L}(f_{\theta'}; x, y) \right\| \leq L_{z\theta} \, \|\theta - \theta'\|.$$

**Lemma A.5** (Smoothness of the penalized surrogate). *Suppose the loss $\mathcal{L} : \Theta \times (\mathcal{X} \times \mathcal{Y}) \to \mathbb{R}$ satisfies Assumptions A.1 and A.3 (smoothness in $\theta$ and $z$) with constants $L_{\theta\theta}, L_{\theta z}, L_{z\theta}, L_{zz}$, and that the transport cost $c(z, w)$ is convex in $w$. Let $F_g$ denote the robust loss for each environment such that*

$$F_g(\theta, x, y) = \sup_{(x', y') \in (\mathcal{X}, \mathcal{Y})} \phi(f_\theta; (x, y), (x', y')),$$

*where*

$$\phi(f_\theta; (x, y), (x', y')) = \mathcal{L}(f_\theta; x', y') - \gamma c((x, y), (x', y')).$$

*Then we have that $F_g$ is $L_f-$smooth, where*

$$L_f = L_{\theta\theta} + \frac{L_{\theta z} L_{z\theta}}{[\gamma - L_{zz}]_+}.$$

*Proof.* The proof follows from Lemma 1 in Sinha et al. (2017) $\qquad\square$

**Lemma A.6** (Lipschitzness of the robust surrogate). *Under Assumption A.2, the robust group-level loss*

$$F_g(\theta, x, y) = \sup_{(u,v) \in \mathcal{X} \times \mathcal{Y}} \left\{ \mathcal{L}(f_\theta; u, v) - \gamma \, c((x, y), (u, v)) \right\}$$

*is K-Lipschitz in $\theta$.*

*Proof.* Let

$$F_g(\theta; x, y) = \sup_{(u,v) \in \mathcal{X} \times \mathcal{Y}} \left\{ \mathcal{L}(f_\theta; u, v) - \gamma \, c((x, y), (u, v)) \right\}.$$

**(i)** $F_g(\theta; x, y) - F_g(\theta'; x, y) \leq K \, \|\theta - \theta'\|$.

Choose $(u^*, v^*) \in \arg\max_{(u,v)}\{\mathcal{L}(f_\theta; u, v) - \gamma \, c((x, y), (u, v))\}$, which according to Sinha et al. (2017) exists for $\gamma > L_{zz}$. Then

$$F_g(\theta; x, y) = \mathcal{L}(f_\theta; u^*, v^*) - \gamma \, c((x, y), (u^*, v^*)),$$

and by definition of the supremum,

$$F_g(\theta'; x, y) \geq \mathcal{L}(f_{\theta'}; u^*, v^*) - \gamma \, c((x, y), (u^*, v^*)).$$

Subtracting gives

$$F_g(\theta; x, y) - F_g(\theta'; x, y) \leq \mathcal{L}(f_\theta; u^*, v^*) - \mathcal{L}(f_{\theta'}; u^*, v^*).$$

By Assumption A.2,

$$\left| \mathcal{L}(f_\theta; u^*, v^*) - \mathcal{L}(f_{\theta'}; u^*, v^*) \right| \leq K \, \|\theta - \theta'\|.$$

Hence

$$F_g(\theta; x, y) - F_g(\theta'; x, y) \leq K \, \|\theta - \theta'\|.$$

**(ii)** $F_g(\theta'; x, y) - F_g(\theta; x, y) \le K \|\theta - \theta'\|$.

Choose $(\tilde{u}, \tilde{v}) \in \arg\max_{(u,v)} \{\mathcal{L}(f_{\theta'}; u, v) - \gamma\, c((x, y), (u, v))\}$. Then

$$F_g(\theta'; x, y) = \mathcal{L}(f_{\theta'}; \tilde{u}, \tilde{v}) - \gamma\, c((x, y), (\tilde{u}, \tilde{v})),$$

and

$$F_g(\theta; x, y) \ge \mathcal{L}(f_\theta; \tilde{u}, \tilde{v}) - \gamma\, c((x, y), (\tilde{u}, \tilde{v})).$$

Subtracting yields

$$F_g(\theta'; x, y) - F_g(\theta; x, y) \le \mathcal{L}(f_{\theta'}; \tilde{u}, \tilde{v}) - \mathcal{L}(f_\theta; \tilde{u}, \tilde{v}) \le K \|\theta - \theta'\|.$$

Combining (i) and (ii) gives

$$\left| F_g(\theta; x, y) - F_g(\theta'; x, y) \right| \le K \|\theta - \theta'\|.$$

$\square$

### A.3  Convergence Analysis for Descent–Mirror-Ascent

From Lemmas A.5 and A.6 we have that the robust loss per group

$$F_g(\theta) = \mathbb{E}_{(x,y)\sim\mathbb{P}^g_{X,Y}} \sup_{(u,v)\in(\mathcal{X},\mathcal{Y})} \{\mathcal{L}(f_\theta; u, v) - \gamma c((x, y), (u, v))\}$$

is $L_f$−smooth and $K$−Lipschitz. Then we have the following lemma.

**Lemma A.7.** *Suppose for each group $g = 1, \dots, G$ the robust-loss function*

$$F_g(\theta) = \mathbb{E}_{(x,y)\sim\mathbb{P}^g_{X,Y}} \sup_{(u,v)\in\mathcal{X}\times\mathcal{Y}} \{\mathcal{L}(f_\theta; u, v) - \gamma\, c((x, y), (u, v))\}$$

*is*

- *$\ell$-smooth in $\theta$: $\|\nabla F_g(\theta) - \nabla F_g(\theta')\| \le \ell \|\theta - \theta'\|$, and*

- *$K$-Lipschitz in $\theta$: $\left| F_g(\theta) - F_g(\theta') \right| \le K \|\theta - \theta'\|$.*

*Define the weighted aggregate*

$$\Psi(\theta, q) = \sum_{g=1}^{G} q_g F_g(\theta), \qquad (\theta, q) \in \Theta \times \Delta_G.$$

*Then:*

1. *$\Psi$ is $\ell$-smooth and*

2. *$\Psi(\theta, \cdot)$ is $K$-Lipschitz in $\theta$, uniformly over $q \in \Delta_G$.*

*Proof.* Fix any $q \in \Delta_G$ and $\theta, \theta' \in \Theta$. Since the weights $q_g \ge 0$ sum to 1, we have

$$\|\nabla_\theta \Psi(\theta, q) - \nabla_\theta \Psi(\theta', q)\| = \left\| \sum_{g=1}^{G} q_g (\nabla F_g(\theta) - \nabla F_g(\theta')) \right\| \le \sum_{g=1}^{G} q_g \ell \|\theta - \theta'\|$$

$$= \ell \|\theta - \theta'\|.$$

Thus $\Psi(\cdot, q)$ is $\ell$-smooth. Similarly

$$\|\nabla_q \Psi(\theta, q) - \nabla_q \Psi(\theta, q')\| = 0 \le 0 \|q - q'\|.$$

$\Psi$ is $\max\{0, \ell\} = \ell$-smooth. Likewise

$$\left| \Psi(\theta, q) - \Psi(\theta', q) \right| = \left| \sum_{g=1}^{G} q_g (F_g(\theta) - F_g(\theta')) \right| \le \sum_{g=1}^{G} q_g K \|\theta - \theta'\| = K \|\theta - \theta'\|.$$

So $\Psi(\cdot, q)$ is $K$-Lipschitz in $\theta$. $\square$

We define

$$\Psi(\theta, q) = \sum_{g=1}^{G} q_g F_g(\theta).$$

and from Lemma A.7 we know that it is $L_f$−smooth and $K$−Lipschitz.

We also define

$$P(\theta) = \max_{q \in \Delta_G} \Psi(\theta, q)$$

. In order to prove convergence we use the notion of stationarity based on the Moreau envolope, such as $P_{1/2L_f}(\theta) = \min_{\theta'} P(\theta') + L_f||\theta' - \theta||_2^2$. In this case, showing that the gradient of Moreau envolope converges to a small value is equal to showing that $\theta$ converges to a stationary point as shown in Davis and Drusvyatskiy (2019).

**Lemma A.8.** *Given assumptions A.2 and A.3, let $\Delta_t = P(\theta_t) - \Psi(\theta_t, q_t)$. Then, we have*

$$P_{1/2L_f}(\theta_t,) \leq P_{1/2L_f}(\theta_{t-1}) + 2\eta_\theta L_f \Delta_{t-1} - \frac{\eta_\theta}{4}||\nabla P_{1/2L_f}(\theta_{t-1})||^2 + \eta_\theta L_f K^2$$

*Proof.* The proof follows the same steps as the proof in the GDA version of Lemma D.3 in Lin et al. (2020) □

**Lemma A.9.** *Given assumptions A.2 and A.3, let $\Delta_t = P(\theta_t) - \Psi(\theta_t, q_t)$. Then $\forall s \leq t - 1$ we have*

$$\Delta_{t-1} \leq \eta_\theta K^2 (2t - 2s - 1) + (\Psi(\theta_t, q_t) - \Psi(\theta_{t-1}, q_{t-1})) + L_f \big(D_{KL}(q^\star(\theta_s)||q_{t-1}) - D_{KL}(q^\star(\theta_s)||q_t)\big)$$

*where $q^\star(\theta) = arg\max_{q \in \Delta_G} \Psi(\theta, q)$.*

*Proof.* By the definition of Bregman Divergence we have that

$$\eta_q(q - q_t)^T \nabla_q \Psi(\theta_{t-1}, q_{t-1}) \leq D_{KL}(q||q_{t-1}) - D_{KL}(q||q_t) - D_{KL}(q_t||q_{t-1}).$$

Since $\Psi(\theta_{t-1}, \cdot)$ is concave we have

$$\Psi(\theta_{t-1}, q) \leq \Psi(\theta_{t-1}, q_{t-1}) + (q - q_{t-1})\nabla_q \Psi(\theta_{t-1}, q_{t-1}) \quad (1).$$

Since $\Psi(\theta_{t-1}, \cdot)$ is $L_f$−smooth we have

$$-\Psi(\theta_{t-1}, q_t) \leq -\Psi(\theta_{t-1}, q_{t-1}) - (q_t - q_{t-1})\nabla_q \Psi(\theta_{t-1}, q_{t-1}) + L_f D_{KL}(q_t||q_{t-1}) \quad (2).$$

Adding (1) and (2), for $\eta_q = \frac{1}{L_f}$ and given the Bregman Definition, we get

$$\Psi(\theta_{t-1}, q) - \Psi(\theta_{t-1}, q_t) \leq (q - q_t)\nabla_q \Psi(\theta_{t-1}, q_{t-1}) + L_f D_{KL}(q_t \| q_{t-1})$$

$$\leq L_f \left[ D_{KL}(q \| q_{t-1}) - D_{KL}(q \| q_t) \right]$$

Plugging $q = q^\star(\theta_s)$ for $s \leq t - 1$ we get

$$\Psi(\theta_{t-1}, q^\star(\theta_s)) - \Psi(\theta_{t-1}, q_t) \leq L_f \left[ D_{KL}(q^\star(\theta_s) \| q_{t-1}) - D_{KL}(q^\star(\theta_s) \| q_t) \right]$$

By the definition of $\Delta_{t-1}$ we have

$$\Delta_{t-1} \leq \big(\Psi(\theta_{t-1}, q^\star(\theta_{t-1})) - \Psi(\theta_{t-1}, q^\star(\theta_s))\big)$$
$$+ \big(\Psi(\theta_t, q_t) - \Psi(\theta_{t-1}, q_{t-1})\big)$$
$$+ L_f \left[ D_{KL}(q^\star(\theta_s) \| q_{t-1}) - D_{KL}(q^\star(\theta_s) \| q_t) \right].$$

Since $\Psi(\theta_s, q^\star(\theta_s)) \geq \Psi(\theta_s, q)$ for all $q \in \Delta_G$, we obtain

$$\Psi(\theta_{t-1}, q^*(\theta_{t-1})) - \Psi(\theta_{t-1}, q^*(\theta_s)) \leq \left[ \Psi(\theta_{t-1}, q^*(\theta_{t-1})) - \Psi(\theta_s, q^*(\theta_{t-1})) \right]$$
$$+ \left[ \Psi(\theta_s, q^*(\theta_s)) - \Psi(\theta_{t-1}, q^*(\theta_s)) \right].$$

Since $\Psi(\cdot, q)$ is $K$–Lipschitz in $\theta$ for any fixed $q$, it follows that

$$\Psi(\theta_{t-1}, q^*(\theta_{t-1})) \; - \; \Psi(\theta_s, q^*(\theta_{t-1})) \;\leq\; K\left\|\theta_{t-1} - \theta_s\right\| \;\leq\; \eta_\theta K^2 (t - 1 - s),$$
$$\Psi(\theta_s, q^*(\theta_s)) \; - \; \Psi(\theta_{t-1}, q^*(\theta_s)) \;\leq\; K\left\|\theta_{t-1} - \theta_s\right\| \;\leq\; \eta_\theta K^2 (t - 1 - s),$$
$$\Psi(\theta_{t-1}, q_t) \; - \; \Psi(\theta_t, q_t) \;\leq\; K\left\|\theta_{t-1} - \theta_t\right\| \;\leq\; \eta_\theta K^2.$$

Putting these pieces together yields the wanted result. $\qquad\square$

**Lemma A.10.** *Given assimptions A.3 and A.2 let $\Delta_t = P(\theta_t) - \Psi(\theta_t, q_t)$. Then the following statement holds*

$$\frac{1}{T+1}\left(\sum_{t=0}^{T} \Delta_t\right) \leq \eta_\theta K^2(B+1) + \frac{L_f D}{B} + \frac{\hat{\Delta}_0}{T+1}$$

*where $\hat{\Delta}_0 = P(\theta_0) - \Psi(\theta_0, q_0)$, $B$ is the block size of how we group the $\Delta_s \forall s \in [0, T]$ and where $D$ is the upper bound of the simplex $\Delta_G$ where $q$ takes values in using the Bregman Divergence.*

*Proof.* In the deterministic setting, we partition the sequence $\{\Delta_t\}_{t=0}^T$ into blocks with size at most $B$:

$$\{\Delta_t\}_{t=0}^{B-1}, \quad \{\Delta_t\}_{t=B}^{2B-1}, \quad \ldots, \quad \{\Delta_t\}_{t=T-B+1}^{T}.$$

There are $\left\lceil (T+1)/B \right\rceil$ such blocks. Hence

$$\frac{1}{T+1}\sum_{t=0}^{T} \Delta_t \;\leq\; \frac{B}{T+1} \sum_{j=0}^{\left\lceil (T+1)/B \right\rceil - 1} \left(\frac{1}{B} \sum_{t=jB}^{\min\{(j+1)B-1, \, T\}} \Delta_t\right).$$

Furthermore, setting $s = 0$ in the inequality of Lemma A.9 yields

$$\sum_{t=0}^{B-1} \Delta_t \;\leq\; \eta_\theta K^2 B^2 \;+\; L_f\left(D_{KL}(q^\star(\theta_s)\|q_{t-1}) - D_{KL}(q^\star(\theta_s)\|q_t)\right) \;+\; (\Psi(\theta_B, q_B) - \Psi(\theta_0, q_0))$$

$$\leq\; \eta_\theta K^2 B^2 \;+\; L_f \; D \;+\; (\Psi(\theta_B, q_B) - \Psi(\theta_0, q_0)).$$

Similarly, letting $s = jB$ for $1 \leq j \leq \left\lceil (T+1)/B \right\rceil - 1$ in Lemma A.9 gives

$$\sum_{t=jB}^{(j+1)B-1} \Delta_t \;\leq\; \eta_\theta K^2 B^2 \;+\; L_f D \;+\; \left[\Psi(\theta_{(j+1)B}, q_{(j+1)B}) \;-\; \Psi(\theta_{jB}, q_{jB})\right].$$

Combining the above gives

$$\frac{1}{T+1}\sum_{t=0}^{T} \Delta_t \;\leq\; \eta_\theta K^2 B \;+\; \frac{L_f D}{B} \;+\; \frac{\Psi(\theta_{T+1}, q_{T+1}) - \Psi(\theta_0, q_0)}{T+1}.$$

Since $\Psi(\cdot, q)$ is $K$-Lipschitz in $\theta$ for any fixed $q$, it follows that

$$\Psi(\theta_{T+1}, q_{T+1}) \;-\; \Psi(\theta_0, q_0) = \left[\Psi(\theta_{T+1}, q_{T+1}) - \Psi(\theta_0, q_{T+1})\right] \;+\; \left[\Psi(\theta_0, q_{T+1}) - \Psi(\theta_0, q_0)\right]$$

$$\leq \eta_\theta K^2 (T+1) \;+\; \hat{\Delta}_0,$$

where $\hat{\Delta}_0 = \Psi(\theta_0, q_0^\star) - \Psi(\theta_0, q_0)$.

$\qquad\square$

**Theorem A.1** (Convergence of Descent–Mirror-Ascent). *Under Assumptions A.2 and A.3, and choosing the step-sizes*

$$\eta_\theta \;=\; \min\left\{\frac{\varepsilon^2}{16 L_f K^2}, \; \frac{\varepsilon^4}{4096 \, L_f^3 K^2 D}\right\}, \qquad \eta_q \;=\; \frac{1}{L_f},$$

*Algorithm 2 returns an $\varepsilon$-stationary point of*

$$P(\theta) \;=\; \max_{q \in \Delta_G} \sum_{g=1}^{G} q_g \, F_g(\theta)$$

*in at most*

$$\mathcal{O}\Big( \frac{L_f^3 K^2 D \, \hat{\Delta}_P}{\varepsilon^6} \;+\; \frac{L_f^3 D \, \hat{\Delta}_0}{\varepsilon^4} \Big)$$

*iterations* $\hat{\Delta}_0 = P(\theta_0) - \Psi(\theta_0, q_0)$ *and* $\hat{\Delta}_P = P_{1/2L_f}(\theta_0) - \min_\theta P_{1/2L_f}(\theta)$.

*Proof.* The proof follows the same steps to combine lemmas A.8, A.9, and A.10 as in Theorem ... in Sinha et al. (2017). The only difference is that we define $B$ as $B = \frac{D}{K} \sqrt{\frac{L_f}{\eta_\theta}}$. $\qquad\square$

## B  EXPERIMENTAL DETAILS

### B.1  DATASETS

#### B.1.1  ADULT INCOME DATASET

We use the UCI Adult dataset Becker and Kohavi (1996), which contains demographic and occupational information for 47,621 individuals, along with a binary label indicating whether annual income exceeds \$50,000.

**Preprocessing.**  We begin by fetching the dataset from the UCI repository and removing rows with missing values. Some race categories (*Amer-Indian-Eskimo*, *Asian-Pac-Islander*) are merged into a single *Other* category to simplify group definitions. All categorical features are encoded using `LabelEncoder`, and continuous features are normalized with `StandardScaler`. The target is binarized as 1 for $> \$50,000$ and 0 for $\leq \$50,000$.

**Group Definitions.**  We construct groups by crossing the sensitive attribute race with the income label. This yields six groups. The distribution of individuals across these groups is shown below:

- **Group 0** — *White, income $>50K$*: 10,485 individuals ($\approx 22\%$).
- **Group 1** — *White, income $\leq 50K$*: 30,301 individuals ($\approx 64\%$).
- **Group 2** — *Black, income $>50K$*: 555 individuals ($\approx 1\%$).
- **Group 3** — *Black, income $\leq 50K$*: 3,980 individuals ($\approx 8\%$).
- **Group 4** — *Other race, income $>50K$*: 501 individuals ($\approx 1\%$).
- **Group 5** — *Other race, income $\leq 50K$*: 1,799 individuals ($\approx 4\%$).

**Train/Test Splits.**  For each of ten seeds, we split the dataset into train and test sets with a ratio of 70–30, stratified by group. For robustness evaluation, we induce a covariate shift by enforcing a *uniform distribution* over the attribute `education` in the training set, while leaving the test distribution as in the original dataset. This ensures a mismatch between training and test environments, which serves as a controlled distributional shift.

**Distributions.**  Figures 5 and 6 show the distributions over `education` for sections 4.1 and 4.2 respectively. These illustrate the impact of our preprocessing and the induced train–test shift.

#### B.1.2  STROKE DATASET

We use the public healthcare stroke dataset, which includes demographic and clinical covariates and a binary label indicating stroke occurrence.

**Preprocessing.**  We remove rows with missing values and drop entries with `smoking_status=`*Unknown*. Smoking is binarized as `never smoked`$\to 0$ and {`formerly smoked, smokes`}$\to 1$. Categorical variables are encoded as follows: `gender` $\in$ {Male, Female, Other} $\to \{0, 1, 2\}$, `ever_married` (No/Yes)$\to \{0, 1\}$, `work_type` via categorical codes, and `Residence_type` (Rural/Urban)$\to \{0, 1\}$. We define the prediction target `target=stroke` and drop `heart_disease`, `stroke`, and `id`. All remaining features are scaled to $[-1, 1]$ with `MinMaxScaler`.

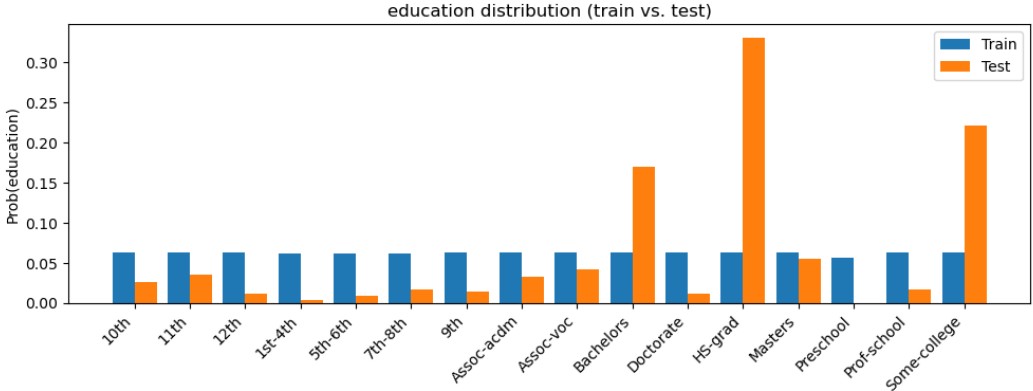

**Figure 5:** Train–test shift in the `education` marginal on the Adult dataset. We construct training splits with a uniform distribution over `education`, while the test split retains the dataset's natural distribution. The example shown is for seed 42; other seeds ( 18, 1999, 2025, etc.) realize the same pattern by construction.

**Group Definitions.** Groups are the Cartesian product of an age bin and the outcome: `age_bin=`$\mathbb{1}\{$`age` $\geq 60\}$ crossed with `target` (stroke/no stroke), yielding four groups:

- **Group 0**: age $< 60$, stroke$= 0$: 2,324 individuals ($\approx 67.8\%$).

- **Group 1**: age $< 60$, stroke$= 1$: 48 individuals ($\approx 1.4\%$).

- **Group 2**: age $\geq 60$, stroke$= 0$: 922 individuals ($\approx 26.9\%$).

- **Group 3**: age $\geq 60$, stroke$= 1$: 132 individuals ($\approx 3.9\%$).

**Train/Test Splits (Smoking environments).** For each of ten seeds, we construct a training set by sampling *equal* numbers from the two smoking categories and shuffling (after scaling, `smoking_status`$\in \{-1, 1\}$), and we use the remaining samples as the test set. This induces a controlled mismatch between the training distribution (balanced by smoking) and the test distribution (natural). We emit a warning if any group is missing in either split.

**Distributions.** Figures 7 and 8 report the distributions of `smoking_status` for sections 4.1 and 4.2 respectively. These illustrate the impact of our preprocessing and the induced train–test shift.

### B.1.3 Colored MNIST

**Base dataset and label binarization.** We start from the torchvision MNIST dataset (`train=True/False`). Each example's digit $d \in \{0, \ldots, 9\}$ is mapped to a binary label. In particular for digits 0–5 $y = 0$, and for digits 6–9 $y = 1$.

**Digit-marginal control (environment shift).** To induce a controlled change in the digit distribution between train and test, we subsample each split using per-digit inclusion probabilities. By default, the training split keeps 80% of digits $\{0, 1, 2, 5, 6, 7\}$ and 20% of digits $\{3, 4, 8, 9\}$, while the test split swaps these rates (keeps 20% of $\{0, 1, 2, 5, 6, 7\}$ and 80% of $\{3, 4, 8, 9\}$).

**Color assignment (spurious correlation).** For each included image, color is coupled to the binary label in training but neutral in testing. *Train:* with probability 0.9 we assign *red* if $y_{\text{bin}} = 0$ and *green* if $y_{\text{bin}} = 1$; with probability 0.1 we flip the color. *Test:* we use a 50–50 split independent of the label, i.e., $\Pr(\text{red} \mid y_{\text{bin}}) = \Pr(\text{green} \mid y_{\text{bin}}) = 0.5$ for both $y_{\text{bin}} \in \{0, 1\}$. Coloring is implemented by copying the grayscale MNIST intensities into a single RGB channel (red channel for "red", green channel for "green") and setting the other channels to zero. The final digits resemble those in Fig. 9.

**Groups.** Groups are defined as the cartesian product of color and binary label: `red_0`, `green_0`, `red_1`, `green_1`.

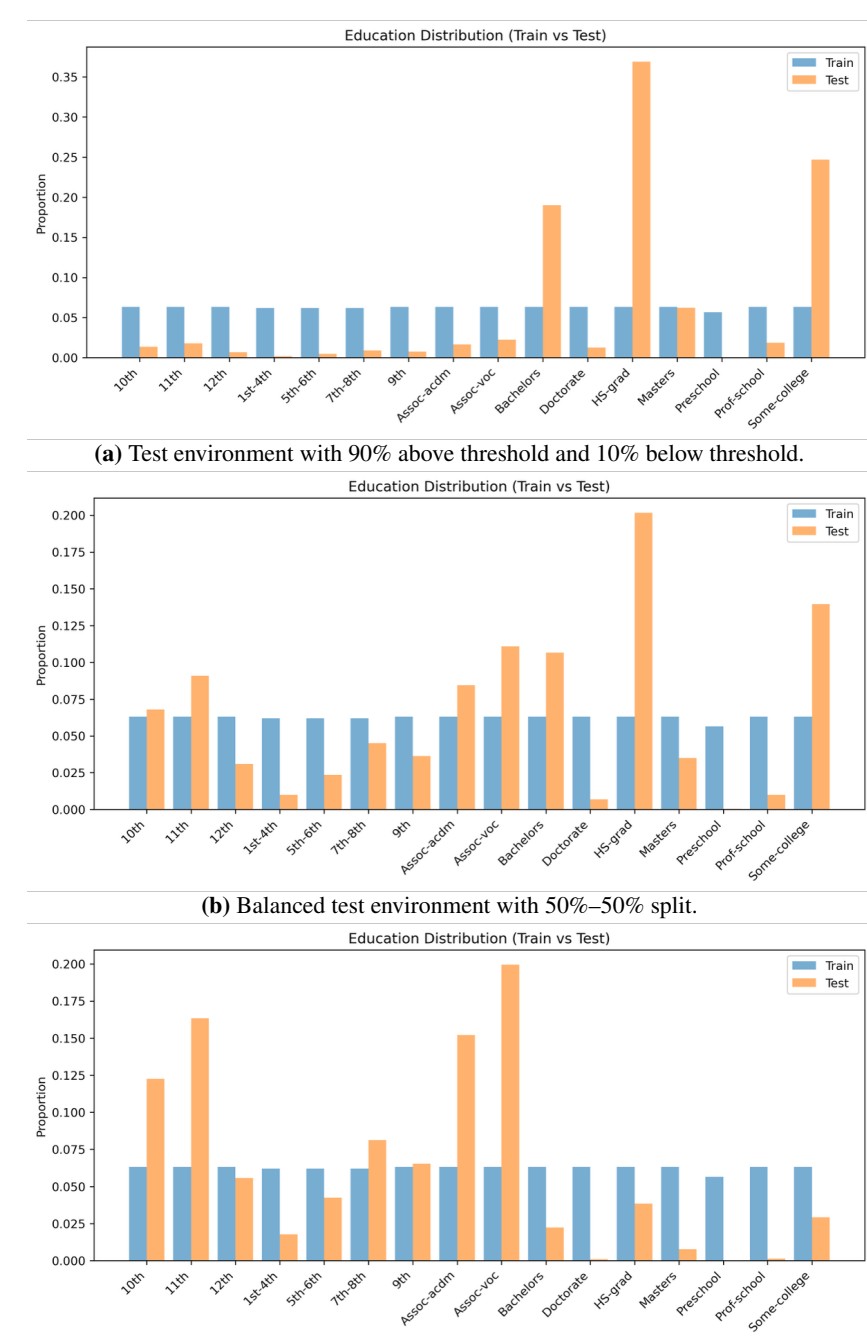

**(a)** Test environment with 90% above threshold and 10% below threshold.

**(b)** Balanced test environment with 50%–50% split.

**(c)** Test environment with 10% above threshold and 90% below threshold.

**Figure 6:** Examples of constructed test environments by varying the proportion of samples above vs. below the education threshold. Shown are the extreme settings (90–10, 10–90) and the balanced case (50–50).

## B.2    MODEL ARCHITECTURES

**Tabular models.**    For all tabular datasets, we employ a multilayer perceptron (MLP) with two hidden layers. The first hidden layer maps the input features to a 64-dimensional representation, followed by an Exponential Linear Unit (ELU) activation. The second hidden layer consists of 32 units, also followed by an ELU activation. The output layer is a single linear unit producing a scalar prediction. Unless otherwise stated, no dropout is applied. This architecture is kept fixed across all methods to ensure fair comparisons.

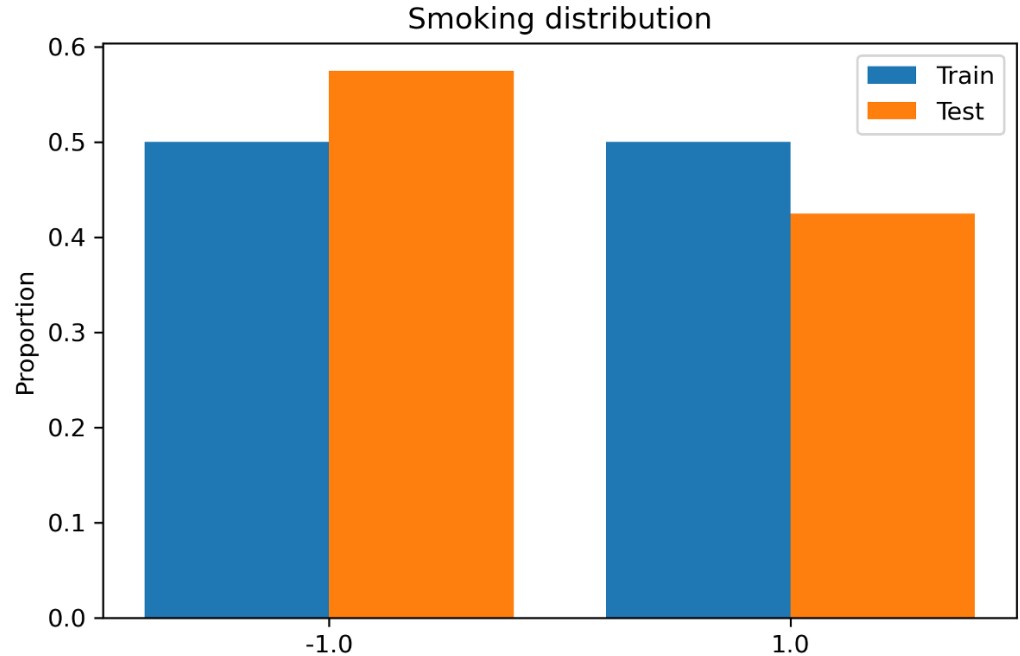

**Figure 7:** Train–test shift in the `smoking` marginal on the Stroke dataset. We construct training splits with a uniform distribution over `smoking`, while the test split retains the dataset's natural distribution. The example shown is for seed 42; other seeds ( 18, 1999, 2025, etc.) realize the same pattern by construction.

**Colored MNIST.** For the Colored MNIST experiments, we use a simple convolutional neural network (CNN) commonly employed in prior work on distributionally robust learning. The network consists of two convolutional layers: the first with 32 filters of size $3 \times 3$, and the second with 64 filters of size $3 \times 3$. Each convolutional layer is followed by a ReLU activation and $2 \times 2$ max pooling. The resulting feature maps are flattened and passed through a fully connected layer with 128 units and ReLU activation, followed by a final linear layer producing a single scalar output. This lightweight CNN is sufficient for the binary classification task (digit $\leq 4$ vs. $> 4$) while allowing controlled evaluation under distribution shifts.

### B.3 HARDWARE AND IMPLEMENTATION

All experiments were implemented in PyTorch (v2.2.0) with CUDA 12.3 and cuDNN enabled. Training and evaluation were performed on a Linux workstation equipped with two NVIDIA TITAN RTX GPUs (24 GB memory each, driver 545.23.08). [1]

## C MULTI-MODAL EXPERIMENTS

### C.1 CHEXPERT (PNEUMONIA, IMAGE+METADATA)

**Dataset and task.** We use frontal chest radiographs from CheXpert to predict pneumonia (binary label). Each example includes an image and structured metadata; in our setup we use `Sex`, standardized age (`Age_z`), and the radiographic finding `Cardiomegaly`, an abnormal enlargement of the heart visible on chest X-rays.

**Preprocessing.** Images are resized to $224 \times 224$ and normalized with ImageNet statistics. Metadata are processed as follows: `Sex` is binarized, `Age_z` is $z$-scored using training statistics, and `Cardiomegaly` is encoded as $\{0, 1\}$.

---

[1]We will release all code and scripts to reproduce our results.

**Groups.** Following the tabular experiments, groups are defined by age and outcome: (age $< 60$ vs. $\geq 60$) $\times$ (pneumonia $0/1$), yielding four intersectional groups.

**Train/test construction (environment shift).** To induce a covariate shift, the *training* split is constructed to have a *uniform* marginal over `Cardiomegaly` (approximately 50–50 present/absent), while the *test* split retains the dataset's natural (imbalanced) prevalence. This mirrors the single-environment shifts used in the tabular studies but on the metadata side of the multimodal input.

**Model (multimodal fusion).** For the CheXpert pneumonia experiments we use a multimodal network that fuses image features with structured metadata (e.g., `Sex`, `Age`). The image backbone is a ResNet 18 with the final fully connected layer replaced by an identity, yielding a global feature vector of dimension `feat_dim` ($= 512$ for ResNet18 ). This vector is projected to `img_out_dim`$=256$ via a linear layer, ReLU, and dropout ($p=0.1$).

Metadata is passed through a small MLP with hidden sizes $\{64\}$ (Linear–ReLU–Dropout). The projected image embedding and the metadata embedding are concatenated and fed to a fusion MLP with hidden sizes $\{128\}$, ending in a single linear logit for binary classification.

**Robustness setup (metadata-only DRO).** In all CheXpert runs, **distributional robustness is applied only to the metadata**, not to the image pixels. Concretely, the DRO inner maximization perturbs the metadata vector in the standardized space using an $\ell_2$ transport cost; images remain fixed. This targets uncertainty in the tabular attributes (e.g., prevalence shifts in `Cardiomegaly`) while avoiding adversarial image perturbations.

**Hyperparameters.** We run 5 different experiments with fixed seeds. For training, we run 200 epochs. Batch sizes are 256 for ERM/DRO and 64 *per group* for GroupDRO/Combined. Optimizers are SGD with momentum 0.9 and weight decay $10^{-4}$; learning rates are 0.05 (ERM, DRO) and 0.1 (GroupDRO, Combined). Group weights use exponentiated gradients with step size $\eta_\ell=0.1$ and one update per epoch. For robustness, we sweep $\gamma \in \{10^{-4}, 10^{-3}, 10^{-2}, 10^{-1}, 1, 2, 3, 4\}$. The inner ascent (applied *only to metadata*, not images) uses 10 steps and ascent learning rate 0.1. Training runs on a single GPU when available.

**Results.** Figure 10 reports performance on the CheXpert pneumonia task. We observe that our method consistently outperforms the baselines in terms of worst-group accuracy, where gains are especially pronounced across mid-range values of $\gamma$. At the same time, our method significantly reduces the accuracy range (right), indicating more equitable performance across groups, while maintaining competitive average accuracy. Interestingly, for very small $\gamma$, performance deteriorates because the induced ambiguity set is overly large: when groups are not far from each other, this over-smoothing leads to a loss of useful structure. In contrast, at moderate values of $\gamma$, our method achieves clear improvements over Group DRO, suggesting that properly calibrated robustness penalties better capture distributional uncertainty without collapsing group-specific signal. Overall, results indicate that our method achieves both higher subgroup robustness and substantially reduced disparities, while maintaining stable average performance.

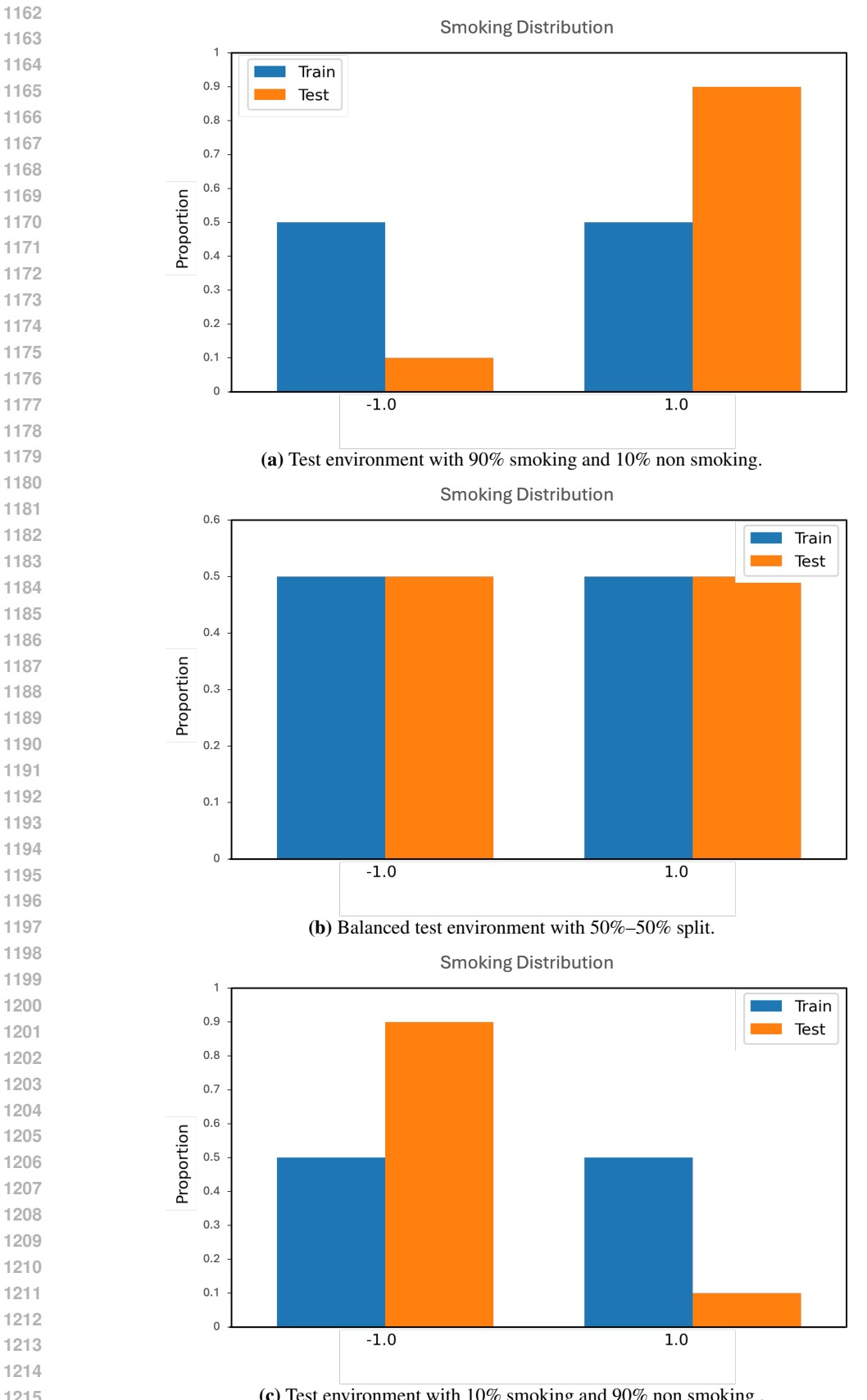

(a) Test environment with 90% smoking and 10% non smoking.

(b) Balanced test environment with 50%–50% split.

(c) Test environment with 10% smoking and 90% non smoking .

**Figure 8:** Examples of constructed test environments by varying the proportion of samples of smoking and non smoking patients. Shown are the extreme settings (90–10, 10–90) and the balanced case (50–50).

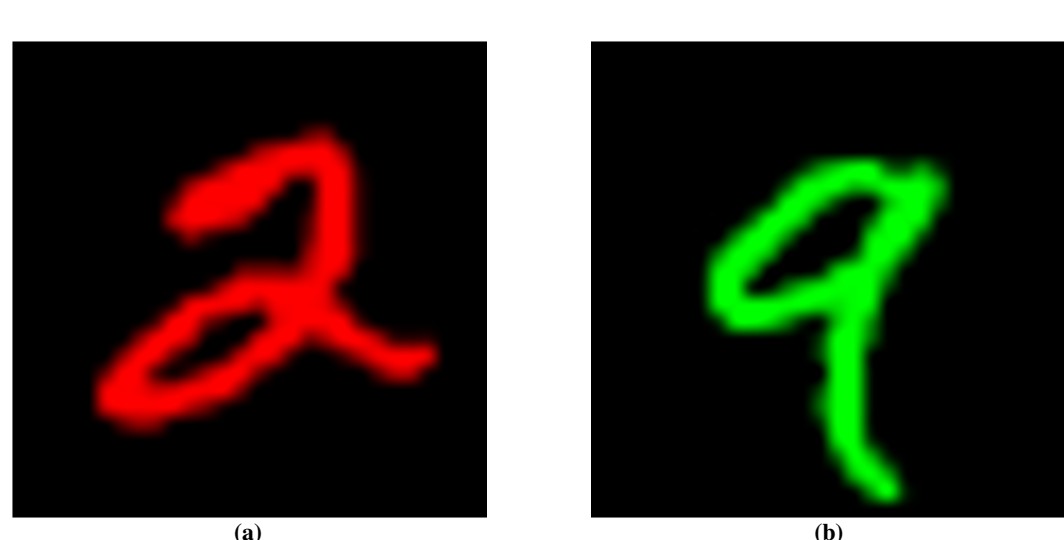

**(a)** **(b)**

**Figure 9:** Two Colored MNIST examples with two different labels and their corresponding colors.

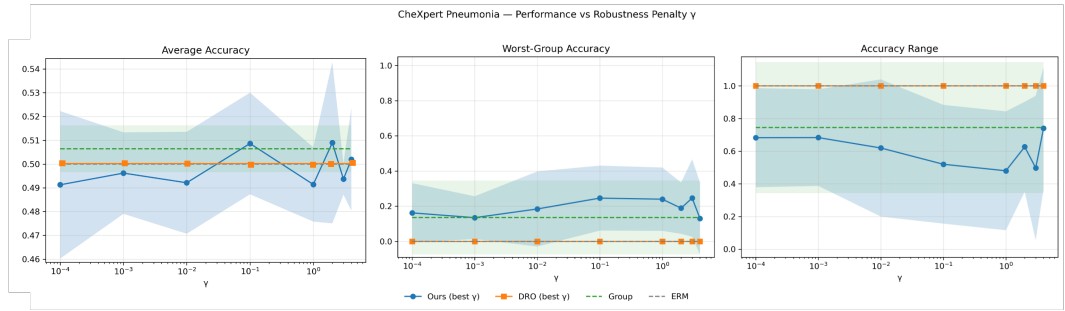

**Figure 10:** Performance of ERM, DRO, Group DRO, and our method on the CheXpert pneumonia prediction task as a function of the robustness penalty $\gamma$. Left: Average accuracy. Middle: Worst-group accuracy. Right: Accuracy range across groups.