# OpenReview forum: "Group Distributionally Robust Machine Learning under Group Level Distributional Uncertainty"
_ICLR.cc/2026/Conference — Submitted to ICLR 2026_

### Official Review · Reviewer_8oPE · 2025-10-27

**Soundness:** 2
**Presentation:** 3
**Contribution:** 3
**Rating:** 4
**Confidence:** 3

**Summary:**

The paper tackles the problem of spurious correlations, which are often learned by models under standard ERM training and yield models that perform well on average but degrade sharply for underrepresented populations.

Traditional approaches include DRO, which models uncertainty in the training distribution and Group DRO, which, in the presence of sub-group annotations, models plausible training distributions on the simplex spanned by the sub-groups and optimizes for the distribution with the worst expected loss.

The caveat of Group DRO is the assumption that the underlying data distributions for each group can be accurately estimated from the training data, without modeling the distributional uncertainty within each group.

This paper focuses on this gap, proposing a method that bridges DRO and Group DRO by modeling the ambiguity set of plausible distributions on the simplex spanned by subgroups and by modeling a per-group ambiguity set as a ball of radius $\epsilon$. The authors claim that their method is thus robust to both changes in the group mixture and to distributional shifts within each group.

Their method operates in three steps:

1. Transform each input samples $(x, y, g)$ into an adversarial example $z$ which maximizes the robust group-level loss.
2. Use the adversarial examples to update the per-group robust losses, their gradients, and update the group weighting parameters $q_g$.
3. Update the model parameters.

This guarantees that each update focuses on the worst-performing plausible distribution within each group and the worst-performing group mixture simultaneously.

Their method is evaluated on tabular and image classification benchmarks, and consistently improves over ERM, DRO, and Group DRO in the following setups:

1. Tabular classification under covariance shift: the training set is sampled such that, for a subset of attributes, their possible values are equally represented, while the test set is left either unchanged or altered to induce a large gap between the likelihoods of potential outcomes (e.g., 50/50 in train, 90/10 in test). Groups are defined using a subset of attributes. Under these benchmarks, their method outperforms ERM, DRO, and Group DRO across all metrics (average and worst-group accuracy, and the accuracy gap between the best- and worst-performing groups).
2. Image classification under spurious correlations and covariate shift: the authors consider the task of predicting whether a number for colored MNIST is < 5 or >= 5. In the training set, the digit color is highly predictive (inducing a spurious correlation). At the same time, this is not the case for the test set. Groups are defined by the (label, color) combination. Furthermore, some digits are over-represented in either the training or test splits. Their method improves the worst-group accuracy by 1-2% over Group DRO, 4% over ERM, and 5% over DRO, respectively, but performs on par with or worse than DRO and ERM in average accuracy.

The authors also investigate a more realistic setup in the appendix using the CheXpert pneumonia classification dataset. While their method performs better on worst-group accuracy, all methods achieve ~50% accuracy (random chance on this binary task). Thus, this benchmark provides limited insights.

**Strengths:**

- The paper is well-written and addresses an important problem.
- The method is novel, theoretically motivated, and shows consistent improvements in worst-group accuracy on tabular classification benchmarks under covariate shifts and on Colored MNIST under spurious correlations.
- The method and experiments are reproducible, as they are clearly documented by the authors in the main paper and appendices.

**Weaknesses:**

- The paper claims to validate the method's effectiveness on "real-world" data. While this might be true for tabular benchmarks, the only meaningful evaluation in vision is on Colored MNIST, as the performance of all methods on CheXpert is close to random (as the classification task is binary) and thus not particularly informative. It would be good to explain this and possibly evaluate the method on other datasets to strengthen the claim.
- The authors do not discuss the scalability of their approach. Since this method produces one adversarial example per input sample and computes gradients for each robust loss, I would expect it to introduce a significant computational overhead compared to DRO or Group DRO. Is this the case?

Other minor details:

- The figures are hard to read without zooming in.
- The Group DRO label in figures is either "Group" or "GDRO". It should be uniform.
- The "100%" at line 274 might be a typo.

**Questions:**

See weaknesses.

---

### Official Review · Reviewer_rnTH · 2025-10-30

**Soundness:** 2
**Presentation:** 1
**Contribution:** 2
**Rating:** 2
**Confidence:** 4

**Summary:**

This papers studies group distributionally robust algorithm. In particular, it incorporates the uncertainty within group nested in group uncertainty while most works only focuses one of those. It develops a three-step gradient algorithm combines adversarial within-group perturbation, group reweighting, and parameter updates and present convergence results. Empirical evaluation on tabular and image datasets compares the method to ERM, classical DRO, and standard GDRO.

**Strengths:**

1) Addressed both group uncertainty and within group uncertainty, which has contributed to address a gap in the literature where most works only consider one of those.

2) The experimental results show significant improvement on worst group accuracy.

**Weaknesses:**

1) The presentation of this paper lacks detailed explanation of the rationale of the proposed formulation and steps of the algorithm. Besides, it did not mention the technical challenges of the propose algorithm and analysis.  For example, Section 3 is not informative at all. Such unclarity not only make the paper hard to read but also leaves the contribution and novelty questionable.

2) The within-group robust loss computation relies on adversarial maximization using gradient ascent (see Algorithm 1), which can become computationally expensive, especially for large scale and high-dimensional problems.

3) It seems unclear about the uncertainty set of the groups, while Wasserstein is used for within group uncertainty. From the perspective, can the inner uncertainty be merged with outer uncertainty so that a standard min-max DRO algorithm is applicable?

4) (Minor) Figure 7 and 8 in the appendix should be made compact since they don't carry much information.

5) (Minor) What is Figure 9 for?

6) (Minor) Line 898 of appendix: theorem index is missed.

**Questions:**

See above.

---

### Official Review · Reviewer_Puxt · 2025-11-01

**Soundness:** 3
**Presentation:** 3
**Contribution:** 2
**Rating:** 2
**Confidence:** 4

**Summary:**

This paper studies the group DRO problem under a group-level worst-case guarantee. There are existing works on (i) group-level worst-case guarantees and existing works on DRO. The paper combines formulations from both sides, as illustrated in Figure 1. The paper proposes a natural gradient descent-ascent algorithm for the minimax formulation and accompanies it with numerical experiments.

**Strengths:**

The paper is well-written and very easy to follow. Audiences with some exposure to distributional robust optimization can easily grasp the ideas and the intuitions behind the formulation and the algorithm.

**Weaknesses:**

My main concerns are:

(i) The formulation is a bit too niche. There is a huge literature on DRO and group-level ML (such as group-level robustness and group-level fairness). Techniques in this paper (including notions of robustness, the algorithm intuitions -- gradient descent/ascent, theoretical analysis of Prop. 3.1)  as noted throughout the paper, are all from the existing literature. I agree that the group DRO + group level worst-case guarantee is an unstudied problem, but from my reading of the paper, I am not convinced that this A+B formulation is a useful one. This double robustness (robust to empirical distribution + robust to worst case group) makes things too conservative.
- Given the progress on self-supervised learning methods such as CLIP from OpenAI has shown other approaches to be more effective in dealing with distributional shift, I don't rate this paper high in that I feel the pursuit along traditional DRO in overcoming distributional shift, if too deep along the path, can be misleading.

(ii) There are three datasets used in this paper. Two tabular and one MNIST-like. It is far from convincing people that the proposed formulation is useful and effective in a more general context ...

**Questions:**

See above.

---

### Official Review · Reviewer_s5yu · 2025-11-05

**Soundness:** 2
**Presentation:** 2
**Contribution:** 2
**Rating:** 4
**Confidence:** 4

**Summary:**

The paper address subpopulation shift problem. The popular solution approach of Group-DRO reweighs the groups during training to focus more on groups with worst loss. However, updating parameters using gradients from worst group is error prone when some groups contain very few examples. This paper tackles the challenge by drawing the worst loss from interpolation regime of $\epsilon$-perturbation of groups, thereby drawing on the advantages of both Group-DRO and DRO. They validate their approach on two tabular datasets and two image datasets.

**Strengths:**

The paper is proposing a well-motivated fix to Group-DRO, which can be suffer from sampling noise in empirical approximation of worst group loss.

**Weaknesses:**

I have a few questions. Will revise this part after rebuttal.

**Questions:**

* Standard datasets are not used. I understand that the standard sub-population shift datasets like WaterBirds, CelebA, MultiNLI, Civilcomments-WILDS are not perfect, but please explain why the paper did not consider them.
* What is the cost function used in experiments as defined in expression (7)?
* As noted in the discussion, I do not see how $\gamma$ interpolates between DRO and G-DRO. Irrespective of $\gamma$ the outer loop still maximizes at group-level unlike DRO.
* Color-MNIST dataset is not the standard one. Worst-group accuracy (of ERM) is far better than usual. Please explain Line 419-420 "To induce a single train-test distribution shift...".
* Can you report on an ablation comparing various methods with decreasing number of examples in the minority group? We should ideally see your approach doing better at very low number of minority examples.

---

### Meta-Review · Area_Chair_Qf11 · 2025-12-23

**Summary:**

The paper proposes a method that bridges Group DRO and distributionally robust optimization (DRO) by modeling both group-level uncertainty and within-group distributional uncertainty. Reviewers generally agree that the problem is well-motivated and relevant, particularly for handling subpopulation shift and spurious correlations. However, opinions diverge on the novelty, usefulness, clarity, and empirical validation of the proposed approach: 1)Limited and unconvincing empirical evaluation; 2) Unclear novelty and practical usefulness; 3) Presentation and clarity issues; and 4) Computational and scalability concerns.

**Reviewer Concerns:**

The authors did not provide any rebuttal.

**Reviewer Scores:**

The authors did not provide any rebuttal.

---

### Decision · Program_Chairs · 2026-01-26

Reject